# Clinico-Pathological Presentations of Cystic and Classic Adenomatoid Odontogenic Tumors

**DOI:** 10.3390/diagnostics10010003

**Published:** 2019-12-20

**Authors:** Primali Rukmal Jayasooriya, Inoka Krishanthi Rambukewella, Wanninayake Mudiyanselage Tilakaratne, Balapuwaduge Ranjit Rigobert Nihal Mendis, Tommaso Lombardi

**Affiliations:** 1Department of Oral Pathology, Faculty of Dental Sciences, University of Peradeniya, Peradeniya 20400, Sri Lanka; primalij@yahoo.com (P.R.J.); inoka_rambukewela@yahoo.com (I.K.R.); 2Department of Oral and Maxillofacial Clinical Sciences, Faculty of Dentistry, University of Malaya, Kuala Lumpur 50603, Malaysia; wmtilak@pdn.ac.lk; 3Laboratory of Oral and Maxillofacial Pathology, Unit of Oral Medicine & Oral Maxillofacial Pathology, University Hospitals of Geneva and Faculty of Medicine, University of Geneva, CUMD, 1206 Geneva, Switzerland; ranjitm@bluewin.ch

**Keywords:** adenomatoid odontogenic tumor, dentinoid, dentigerous cyst, recurrences

## Abstract

The objective of the study is to present the clinico-pathological features of cystic and classic adenomatoid odontogenic tumors (AOTs) in order to identify the differences between the two variants of AOT. Materials and method: The study sample comprised of 41 AOTs, which were categorized into cystic and classic AOTs. Cystic AOTs are diagnosed as such when macroscopic and microscopic evidence of a cyst is present together with histopathological criteria of AOT (WHO–2017). Results: The study sample comprised of eleven cystic and thirty classic AOTs. Eight cystic AOTs were regarded as arising from dentigerous cysts as these lesions were attached to the cemento-enamel junction of the impacted teeth. Though not statistically significant, in contrast to classic AOTs which showed female predilection, cystic AOTs were more prevalent in males. Cystic AOTs tend to present as significantly larger lesions compared to classic AOTs (*p* < 0.02). In both cystic and classic AOTs, duct-like structures and epithelial whorls were the two most prominent histopathological features present in the majority of tumors. Two AOTs with massive amounts of dentinoid occurred in the mandible and presented as large lesions that eroded cortical bone. None of the 12 patients with follow-up information presented with recurrences. Conclusion: Except for the size of the lesion, no significant clinico-pathological differences were observed between cystic and classic AOTs. Therefore the cystic AOTs can be considered as a variant of AOT with enucleation, simple excision, or radical excision as the treatment of choice depending on the extent of the lesion, similar to classic AOTs.

## 1. Introduction

Adenomatoid odontogenic tumor (AOT) is defined as a lesion composed of odontogenic epithelium arranged into a variety of histo-architectural patterns and embedded in mature connective tissue [1]. Although the 2005 WHO blue book on “Pathology and Genetics of Head and Neck Tumours” [1] classifies AOT under the first category containing “tumors composed of odontogenic epithelium only”, the prior WHO classification [2] included the lesion in the second category of “tumors containing both odontogenic epithelium as well as ectomesenchyme with or without dental hard tissues”. Initially, dental matrix material in the form of enameloid and dentinoid present in some AOTs was considered to be a metaplastic change and not true inductive change [1,3]. However, though experts in the field consider the secretory activity of the AOT leading to enameloid and dentinoid formation as a true inductive change [3], and discussions took place to shift AOTs with inductive changes to the second category leaving the remaining lesions in the first category, no such change has been made in the most recent edition of the WHO classification released in 2017 [4].

Even though there is evidence to indicate awareness of the cystic AOT from the early years supported by the fact that cystic presentation has been mentioned in the previous WHO definitions of AOT [2], detailed analyses of these lesions did not take place until recently [5,6]. As such, another reason for the current interest in the AOT is due to the recent publications that deal with cystic tumors [5,6].

Therefore the aim of the present study is to present the clinico-pathological features of cystic and classic (conventional) AOTs in order to identify differences in the two variants.

## 2. Materials and Method

The study sample comprised 41 AOTs diagnosed over a 14 year period from 1999–2013 at the Dept. of Oral Pathology, Faculty of Dental Sciences, University of Peradeniya, Sri Lanka. Ethical clearance for the study was obtained by the Ethical review committee of the Faculty of Dental Sciences (FDS/2014/07, 27 November 2014). All lesions were categorized into two basic groups, namely, cystic and classic AOTs depending on the macroscopic evidence and histopathology. The radiological presentation of follicular and extra-follicular pattern, as described in the 2005 WHO, was not considered when grouping lesions into cystic and classic types. The cystic lesions (Figure 1a) that were completely or partially lined by thin non-keratinized stratified squamous/cuboidal epithelium of 2–3 cell layer thickness with histopathological evidence of AOT were identified as cystic AOTs (Figure 1b), while the remaining lesions were considered as classic AOTs. The cystic lesions were further divided into two groups, with the predominant group of AOT arising in dentigerous cysts containing lesions attached to the cemento-enamel junction of impacted teeth. The remaining cystic AOTs were identified as AOTs arising in unclassifiable odontogenic cysts when the true dentigerous relationship is absent or due to absence of an impacted tooth within the lesion.

Clinical information such as age, gender, site, size, and presence/absence of impacted teeth and recurrences were obtained from the patient’s records and biopsy request forms. Further histopathological characteristics of the AOTs were noted for the two groups of AOTs. However, it was not possible to obtain follow-up information for patients, who were treated prior to 2009, and thus follow-up information was available for 12 patients only. The clinico-pathological information thus obtained were statistically analyzed for cystic and classic AOTs using the X^2^ test (*p* = 0.05). In addition, 09 CEOTs present in the archives of the Department of Oral Pathology were included for the comparison between classic AOTs, AOTs containing CEOT-like areas, and CEOTs.

## 3. Results

The sample comprised of 11 and 30 cystic and classic AOTs, respectively (Table 1). Within the group of cystic AOTs, eight were classified as arising from dentigerous cysts, while the remaining three lesions were considered to arise from unclassifiable odontogenic cysts. Clinico-pathological comparison between cystic and classic AOT is given in Table 1. Accordingly, there were no statistically significant differences in the age and site distribution when the two groups were considered (*p* > 0.7). In contrast, though not statistically significant, higher numbers of cystic tumors were found in males compared to classic AOTs (*p* > 0.25). Further, cystic tumors were larger in size than the classic AOTs (*p* < 0.02).

Although the majority of tumors in both groups were encapsulated lesions (Table 1), a few solid AOTs were un-encapsulated and were seen infiltrating into adjacent bone. It is also worthwhile to mention that one cystic AOT associated with an ossifying fibroma was observed to erode into the maxillary antrum as well. Further, three mandibular AOTs were extensive lesions that crossed the midline. With reference to the type of impacted tooth observed, both cystic and solid AOTs were more commonly associated with impacted canines (13/22) followed by premolars (4/22), incisors (4/22) and second molars (1/22). None of the lesions included in the study sample were associated with impacted third molars.

Out of the 12 patients who were followed-up for 5 years to 9 months, no recurrences were identified. However, as the department is the only institution that deals with oral biopsies in Sri Lanka, it can be concluded that none of the AOTs have recurred, as we have not received any biopsies of recurrent AOTs.

Both cystic and solid AOTs were histopathologically characterized by the presence of duct-like structures, epithelial whorls, and rosette-like structures (Figure 2a). However, though statistically insignificant epithelial whorls were more consistently identified in AOTs compared to duct-like structures (*p* > 0.8) (Table 1). Further, the majority of the tumors presented with tumor droplets and calcifications (Figure 2b). The only unique histopathological feature that differentiated cystic from classic tumors was the complete or partial cystic epithelial lining in the former lesion, while none of the histopathological features of AOT could be used to differentiate the two groups of lesions. Another interesting finding was the presence of dentinoid in AOTs. Out of the two lesions with massive amounts of dentinoid, one tumor presented as a large mandibular lesion which extended from right first premolar to left second premolar region in a 15-year-old female. Histopathologically a predominantly cystic lesion lined by thin bi-layered epithelium with duct-like structures and epithelial whorls were identified, together with large areas of dentinoid/osteodentine and melanin (Figure 3a,b). The other tumor with dentinoid occurred in a 33-year-old female and presented as an un-encapsulated 4 × 4 cm lesion in relation to the lower molars. The tumor was composed of predominantly small blue cells showing duct-like structures, occasional mitoses, and marked osteodentine (Figure 4). Due to the infiltrative nature, both lesions were radically treated with marginal mandibulectomy, in contrast to simple excision or enucleation performed for the majority of the AOTs.

Immunohistochemical investigations with CK19 revealed strong positivity, which confirmed the odontogenic nature of the AOTs (Figure 5a). In addition, Ki-67 staining revealed positivity in approximately 1% of the tumor cells (Figure 5b). This finding can be used to support the non-aggressive behavior of the AOTs.

Table 2 shows the demographic features of cystic AOT of the present series, with a comparison involving published cystic AOTs. Accordingly, it is noteworthy to mention that the AOTs arising from dentigerous cysts show a male predilection (X^2^ test *p* < 0.05). Furthermore, though the sample size is too small to make definite conclusions, AOTs arising from unclassifiable odontogenic cysts may be more common in adults in the third decade of life, in contrast to the AOT arising from dentigerous cysts or classic AOTs, which show a predilection to the second decade of life.

Table 3 shows the comparison of hybrid AOT + CEOT with classic AOT and CEOT. Without a doubt, the comparison shows that hybrid AOT + CEOT (Figure 6) as having a demographic profile similar to classic AOT than CEOT. Further, no recurrences have been identified in AOT + CEOTs or classic AOTs while classic CEOTs have presented with recurrences.

## 4. Discussion

Even after a century following the initial report, the AOT remains a widely researched tumor due to its unique biological profile. Harbitz in 1915 and Wohl in 1916 are credited for the first description of AOT, and even at inception, the cystic nature of the AOT was highlighted by the names used for the lesion, “cystic adamantinoma” and “tooth germ cyst” [7]. However, the subsequent analyses of the tumor [8,9,10] did not considered the cystic nature as a major factor until interest was rekindled by a paper on the topic of cystic AOT by Gadewar and Srikant in 2010 [5]. Further, a letter to the editor entitled “Cystic adenomatoid odontogenic tumor: the master of disguise” now highlights the necessity for further analysis in order to assess differences, if any, with reference to treatment and prognosis of cystic vs. classic AOT [6].

There is controversy regarding the naming of the cystic AOT. Accordingly, the term “AOT arising from dentigerous cyst” can be applied when a cystic tumor is identified enclosing the crown of an impacted tooth and attached to the cemento-enamel junction by a thin non keratinized stratified squamous epithelium, producing nodules of AOT in the cyst capsule [5]. In the present study, the majority of the cystic lesions (8/11) fulfilled the above criteria and were identified as cystic AOTs arising from dentigerous cysts. Another group of cystic AOTs was diagnosed as arising from unclassifiable odontogenic cysts when they did not fulfill the criteria necessary for the diagnosis of dentigerous cysts. Although some classic AOTs included in the present series showed focal cystic change, these were not considered as cystic AOTs, due to the absence of complete or partial cyst epithelial lining. In the present study, follicular or extrafollicular presentation is not considered when classifying cystic and classic AOTs.

As previously reported in [8,9,10,11,12], analysis of demographics of the total sample supports the two-thirds phenomenon with the majority of tumors occurring in the second decade of life in females showing a predilection to the maxilla (Table 1). However, in the present sample, the expected two-thirds phenomenon was not observed when the ratio of follicular vs. extra follicular presentation was considered (Table 1).

In the present study sample, 26.8% of the lesions presented as cystic AOTs which arose from odontogenic cysts. In contrast, higher frequencies of cystic AOTs have been identified by Leon et al. (2005) and de Matos et al. (2012) [11,12].

Similar to previous analyses [5,6], the present study also reveals male predilection as the major difference between AOT arising in the dentigerous cyst and classic AOT (Table 2). According to Lo Muzio et al. (2017) [13] analyzing 152 dentigerous cysts in a pediatric population, a male to female ratio of 1.1:1 was revealed. Therefore, it is not surprising that AOTs arising in dentigerous cysts also show a male predilection. This gender distribution is useful to convince the fact that some AOTs may occur in dentigerous cysts.

However, location-wise both cystic and classic AOTs showed a predilection to the canine region in the present study sample (Table 1 and Table 2), while, in contrast, Gadewar and Srikant [5,6] reveal the premolar region as the commonest location for cystic AOT. Another interesting finding is the larger size of the cystic AOTs in contrast to classic AOTs (Table 1). As the majority of odontogenic cysts enlarge by osmosis, this form of growth may apply to cystic AOTs as well, in addition to its neoplastic growth potential resulting in larger lesions.

The AOT is histopathologically characterized by variably sized nodules or interlacing strands of odontogenic epithelium forming duct-like structures, rosettes, and whorls in a stroma containing calcified material, hyaline tumor droplets and loosely arranged connective tissue (Figure 2a,b) [1,2]. No significant differences were observed with reference to the histopathological presentation of the AOT component of cystic vs. classic lesions. In addition, Jivan et al. 2008 [3] describe the occurrence of secretory structures, in which dental matrix material is surrounded by a single layer of tall columnar secretory cells arranged in a circular or duct-like configuration. The PAS-positive dental matrix material identified within the secretory units have been considered as enameloid-like material [3]. A similar presentation was also observed in some of our cases (Figure 2b) and as suggested by Jivan et al. (2008) [3], this fact can be considered as evidence of inductive changes that occur in AOTs.

In the present study sample, some AOTs were found to behave more aggressively producing larger lesions, cortical erosion, and tooth resorption, requiring radical treatment. Two such mandibular lesions contained massive amounts of dentinoid/osteodentine. Although an adenoid ameloblastoma with dentinoid is a possibility, we were unable to conclusively diagnose our lesion as such due to the absence of foci showing ameloblastoma-like features. According to approximately 13 published cases, adenoid ameloblastoma is a rare odontogenic tumor, combining features of both ameloblastoma and AOT [14]. Although the lesion included in the present study sample has not recurred to the best of our knowledge, according to literature, recurrences are a major feature of adenoid ameloblastoma [14].

According to the literature, AOTs have shown CK5, CK14, CK17 and CK19 positivity, which is similar to the cytokeratin profile of dentigerous cyst and oral gingival epithelium. In addition, the non-aggressive nature of the AOT is supported by low expression of proliferative markers such as PCNA and Ki-67 [15]. In accordance with the literature, similar CK 19 expression patterns and low Ki-67 expression were observed in our cases as well.

AOTs of the present study were treated with enucleation, conservative excision, or radical excision, depending on the extent of the lesion. However, none of these lesions recurred. These findings confirm the fact that there are no significant differences between cystic and classic AOTs with reference to biological behavior or prognosis, provided the patient receives appropriate surgical management based on the size and extent of the lesion.

According to the literature, AOTs may rarely produce recurrences [16]. However, Ide et al. (2009) [17] propose that the AOTs that have recurred may, in fact, be adenoid ameloblastoma and not AOTs. Hence, it is important to differentiate adenoid ameloblastoma from classic AOTs histopathologically, as the former lesion behaves as an ameloblastoma and requires more aggressive treatment.

Unlike in classic CEOT, which present with infiltrative nests or cords of tumor epithelium showing nuclear pleomorphism/hyperchromatism, AOT + CEOTs are non infiltrative lesions that lack nuclear atypia. Accordingly, CEOT-like lesions have been shown not to influence the overall biological behavior of the AOT [10]. In addition, in support of this observation, Table 3 shows a similar biological profile for classic AOT and AOT + CEOT in contrast to classic CEOT. Therefore, the presence of CEOT like areas in an otherwise typical AOT should not influence the management, as the lesion can be expected to behave as an AOT in a non-aggressive manner.

In conclusion, the present study supports the existence of cystic AOTs that may arise from dentigerous cysts. However, no significant differences have been identified between cystic vs. classic AOTs except for the size of the lesion, hence the same management strategies may be applied to both entities. Therefore, the sub-classification of AOT into cystic and classic lesions do not serve an additional purpose. However, it is worthwhile to especially mention to the clinicians the AOTs that present with osteodentine/dentinoid as these lesions behave aggressively and may be more prone to recurrences [16].

## Figures and Tables

**Figure 1 diagnostics-10-00003-f001:**
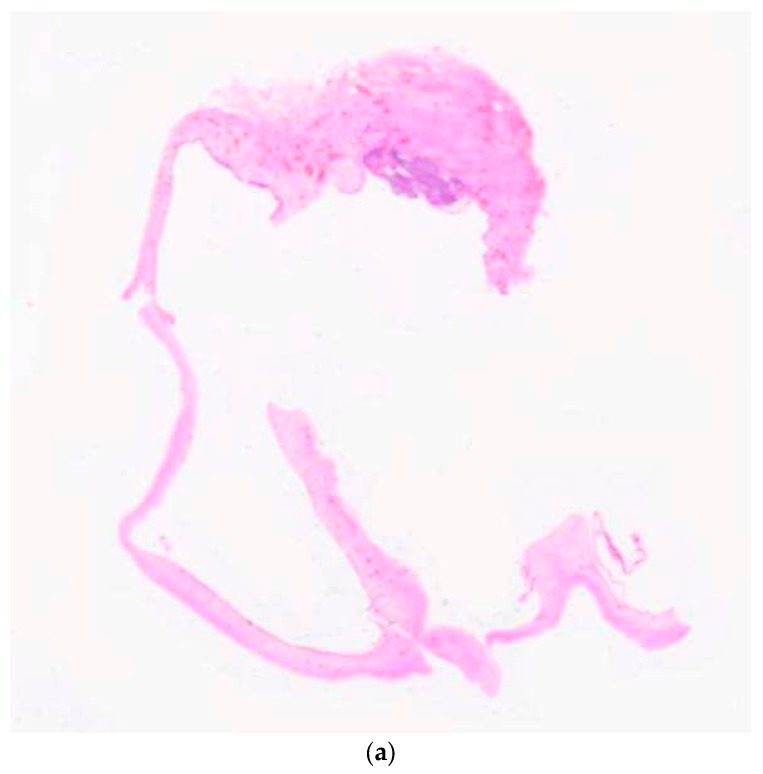
(**a**) Low magnification via scanner only, showing the cystic nature of the lesion. Note: Proliferating tumour mass at the 12 o’clock position. (**b**) Figure showing the entire proliferating mass at the 12 o’clock position, with the thin cyst lining epithelium (×4 H&E—hematoxylin-eosin staining).

**Figure 2 diagnostics-10-00003-f002:**
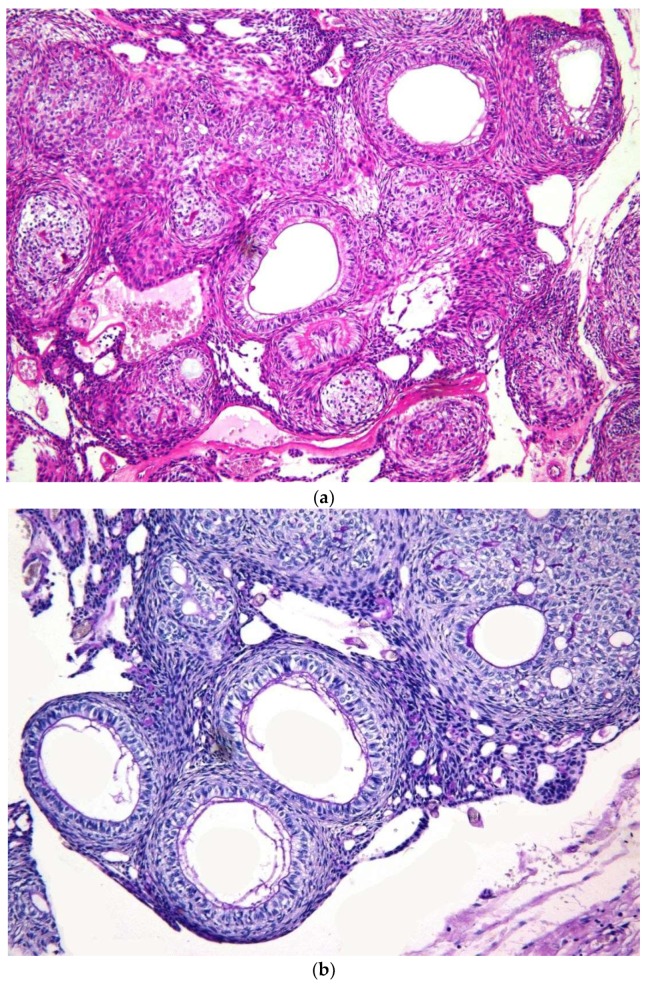
(**a**) The typical histopathological presentation of the epithelial component of AOT. Note the duct-like structures of varying size, with a large secretory duct right in the middle of the picture, followed by a rosette-like structure, showing two epithelial cell layers and eosinophilic hyaline material in between, and immediately beneath it, an epithelial whorl composed of clear cells. (×8 H&E) (**b**) Note the concentration of D-PAS positive material on the inner lumen of the three duct-like structures. The tumor droplets present within the epithelial whorl also show D-PAS positivity (×16, D-PAS).

**Figure 3 diagnostics-10-00003-f003:**
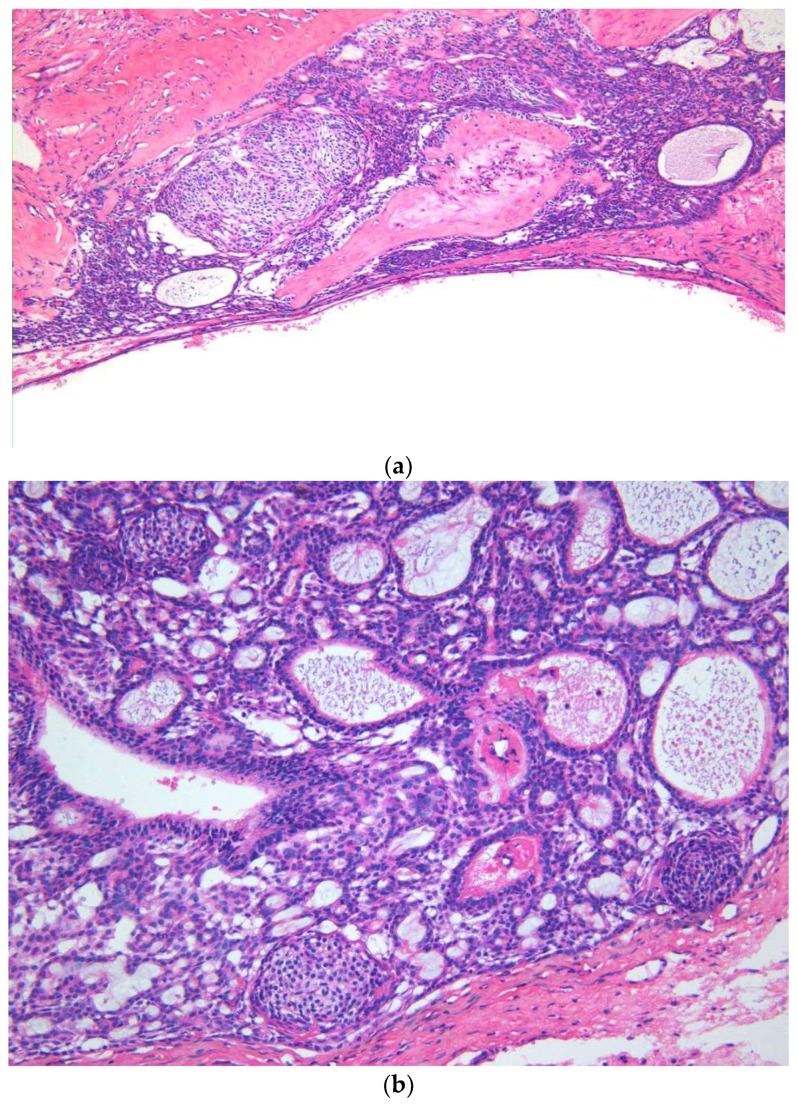
(**a**) Low magnification view showing a predominantly cystic lesion with the epithelial component of AOT and dentinoid/osteodentine in the capsule (×4 H&E). (**b**) The epithelial component of AOT exhibits vague plexiform ameloblastoma-like anastomosing cords of odontogenic epithelium, microcysts, and whorls (×8 H&E).

**Figure 4 diagnostics-10-00003-f004:**
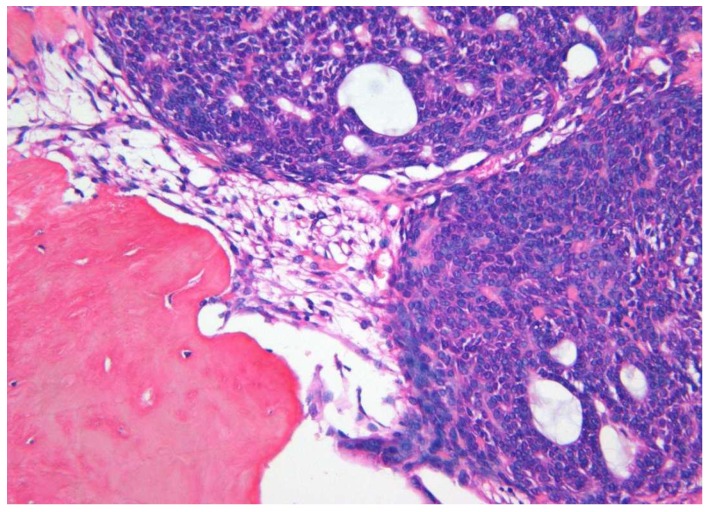
The tumour showing small blue cells with duct like structures and osteodentine (×10 H&E).

**Figure 5 diagnostics-10-00003-f005:**
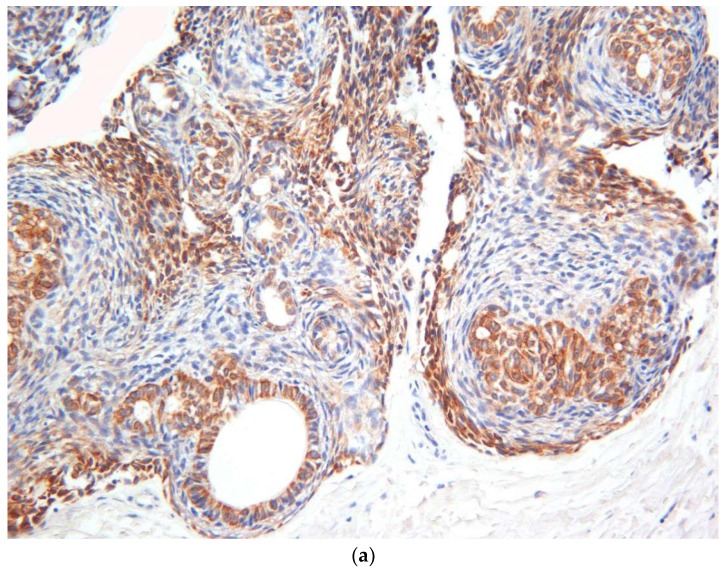
(**a**) Immunohistochemical investigations with CK19 revealed strong focal positivity in the epithelial component of the AOT (×8). (**b**) Immunohistochemical staining with proliferation marker Ki-67 revealed positivity in approximately 1% of the tumor cells (×10).

**Figure 6 diagnostics-10-00003-f006:**
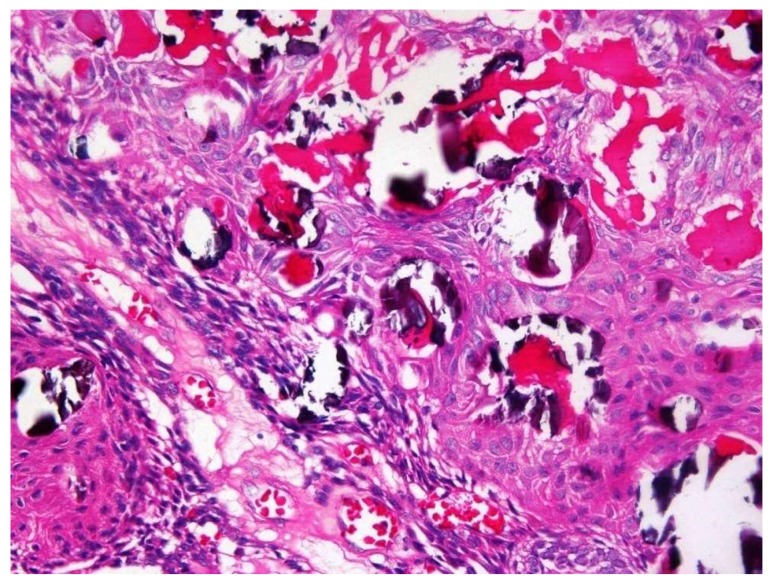
A typical CEOT like area within the AOT showing a collection of polyhedral shaped squamoid epithelial cells with eosinophilic cytoplasms. In contrast to classic CEOT, nuclear atypia is not marked. Pink homogenous amyloid-like material undergoing calcification is also noted (×16 H&E).

**Table 1 diagnostics-10-00003-t001:** Clinico-pathological comparison of adenomatoid odontogenic tumors (AOTs) with cystic and classic presentation.

Clinical Feature	Cystic AOT (%) *n* = 11	Classic AOT (%) *n* = 30	Total (%) *n* = 41	*p* Value
Age				
10–15 yrs	04 (36.4)	11 (36.7)	15	
16–20 yrs	04 (36.4)	15 (50.0)	19	*p* = 0.4
21–25 yrs	00	03 (10.0)	03	
>26 yrs	03 (27.2)	01 (03.3)	04	
Gender				
Female	06 (54.5)	22 (73.3)	28	*p* = 0.3
Male	05 (45.5)	08 (26.7)	13	
Site				
Maxilla	08 (72.8)	19 (63.3)	27	
Mandible	03 (27.2)	10 (33.4)	13	*p* = 0.7
Unknown	00	01 (03.3)	01	
Size				
Less than 3 × 3 cm	01 (09.1)	12 (40.0)	13	
More than 3 × 3 cm	07 (63.6)	08 (26.7)	15	*p* = 0.2
unknown	03 (27.3)	10 (33.3)	13	
Radiological presentation				
Follicular	08 (72.8)	14 (46.7)	22 (53.7)	
Extra follicular	03 (27.2)	14 (46.7)	17 (41.5)	*p* = 0.2
Peripheral	00	01 (03.3)	01 (02.4)	
Unknown	00	01 (03.3)	01 (02.4)	
Histopathology				
1. Capsule-present	10 (90.9)	24 (80.0)	34	
2. Epithelial component				
2a. Duct like structures	09 (81.8)	27 (90.0)	36	
2b. Epithelial whorls	11 (100)	29 (96.6)	40	*p* = 0.7
2c. Rosettes	03 (27.2)	19 (63.3)	22	
2d. Trabeculae	08 (72.8)	25 (83.3)	33	
3. Stromal component				
3a. Tumour droplets	08 (72.8)	20 (66.6)	28	
3b. Calcifications	11 (100)	29 (96.6)	40	*p* = 0.8
3c. Osteo-dentine	01 (09.1)	01 (03.3)	02	
3d. Melanin	00	01 (03.3)	01	
Type of surgery				
Enucleation	10 (90.9)	28 (93.3)	38	*p* = 0.7
Radical surgery	1 (03.3)	2 (06.7)	3	

**Table 2 diagnostics-10-00003-t002:** Demographic features of cystic AOT (comparison of the published series, Srikant (2010), and Grover et al. (2015) with the present cases).

Clinical Features of Cystic AOT	Published Cases		Present Cases (*n* = 11)	
	All cystic AOT (*n* = 19)	AOT arising in dentigerous cysts (*n* = 12)	AOT arising in dentigerous cysts (*n* = 08)	AOT arising in unclassifiable odontogenic cysts (*n* = 03)
Age	Range 0–40 yrs	Range 8–25 yrs	Range 13–27 yrs	Range 18–29yrs
	Average 19.5 yrs	Average 15.5 yrs	Average 16.7yrs	Average 24.3 yrs
Gender	12 out of 19 occurred in males	7 out of 12 occurred in males	5 out of 8 occurred in males	One out of 3 occurred in a male
	Male:female ratio 1.7:1	Male: female ratio 1.4:1	Male: female ratio 1.6:1	Male:female ratio 0.5:1
Site	11 out of 19 occurred in maxilla	11 out of 12 occurred in the maxilla	5 out of 8 cases occurred in the maxilla	All 3 cases occurred in the maxilla
	Maxilla:mandible ratio 1.4:1	Maxilla:mandible ratio 11:1	Maxilla:mandible ratio 1.6:1	
Location according to teeth present	Canine *n* = 9, premolar/molar *n* = 6	Canine *n* = 7, premolar = 3, molar = 2	Incisor *n* = 1, canine *n* = 5, premolar *n* = 1, molar *n* = 1	Incisor *n* = 1, premolar/molar *n* = 2

**Table 3 diagnostics-10-00003-t003:** Comparison of demographic features of classic AOT, AOT with CEOT, and classic CEOT.

Clinical Feature	Classic AOT (*n* = 30)	AOT + CEOT (*n* = 9)	Classic CEOT (*n* = 9)
Age	Range 13–33 yrs	Range 15–25 yrs	Range 26–58 yrs
	Average 18 yrs	Average 17.8 yrs	Average 40 yrs
Gender	22 out of 30 occurred in females	7 out of 9 occurred in females	5 out of 9 occurred in females
	Male: female ratio 1:2.75	Male:female ratio 1: 3.5	Male:female ratio 1: 1.25
Site	19 out of 29 occurred in maxilla	6 out of 9 occurred in maxilla	1 out of 9 occurred in maxilla
	Maxilla:mandible ratio 1.9:1	Maxilla:mandible ratio 2:1	Maxilla:mandible ratio 1:9
Location in the jaw bones	Anterior = 24	Anterior = 7	Anterior = 1
	Premolar/molar = 5	Premolar/molar = 2	Premolar/molar = 5
	Angle of the mandible = 0	Angle of the mandible = 0	Angle of the mandible = 3
Recurrences	None	None	2 out of 9 lesions presented with recurrences within 5 years after treatment

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
