# Peer review of "Clinico-Pathological Presentations of Cystic and Classic Adenomatoid Odontogenic Tumors"

_diagnostics, 2019, doi:10.3390/diagnostics10010003_

Round 1

Reviewer 1 Report

This is an interesting paper, but the reviewer has two issues that need to be clarify:

1) The WHO 2005 say: Intraosseous Adenomatoid odontogenic Tumors (A0T) may be found in association with unerupted permanent teeth (follicular type), Radiographically, the intraosseous, follicular AOT, shows a well-defined, unilocular radiolucency around the crown and often part of the root of an unerupted permanent tooth, mimicking a dentigerous cyst. If not associated with an unerupted tooth (extrafollicular type), AOT presents as a unilocular radiolucent lesion. (WHO 2005).

The "AOT arising from dentigerous cyst " or cystic AOT is the same that follicular type?

Please discuss that point.

2) The authors say: "There is controversy regarding the naming of the cystic AOT. Accordingly, the term “AOT arising from dentigerous cyst” can be applied when a cystic tumour is identified enclosing the crown  of an impacted tooth and attached to the cemento-enamel junction by a thin non keratinized stratified squamous epithelium, producing nodules of AOT in the cyst capsule....

But the WHO 2017 say: AOTs can be radiographically indistinguishable from dentigerous cyst, unless they extends apically beyound the cemento-enamel juntion of the affected tooth (WHO 2017)

Please discuss that point 

Author Response

1 Comments and Suggestions for Authors

This is an interesting paper, but the reviewer has two issues that need to be clarify:

1) The WHO 2005 say: Intraosseous Adenomatoid odontogenic Tumors (A0T) may be found in association with unerupted permanent teeth (follicular type), Radiographically, the intraosseous, follicular AOT, shows a well-defined, unilocular radiolucency around the crown and often part of the root of an unerupted permanent tooth, mimicking a dentigerous cyst. If not associated with an unerupted tooth (extrafollicular type), AOT presents as a unilocular radiolucent lesion. (WHO 2005).

The "AOT arising from dentigerous cyst” or cystic AOT is the same that follicular type?

ANSWER

No. Extrafollicular and follicular presentation described in the 2005 WHO classification is only used to indicate the radiological appearance of the AOT. In our paper, a cystic AOT is diagnosed mainly using the macroscopic and histopathological appearance of the lesion (refer materials method section). Thus the Follicular type described in the AOT can occur in the classic AOT variant as well (refer table 1)

Please discuss that point.

2) The authors say: "There is controversy regarding the naming of the cystic AOT. Accordingly, the term “AOT arising from dentigerous cyst” can be applied when a cystic tumour is identified enclosing the crown of an impacted tooth and attached to the cemento-enamel junction by a thin non keratinized stratified squamous epithelium, producing nodules of AOT in the cyst capsule....

But the WHO 2017 say: AOTs can be radiographically indistinguishable from dentigerous cyst, unless they extends apically beyound the cemento-enamel juntion of the affected tooth (WHO 2017)

ANSWER

As mentioned before, we classified the cystic and classic variants of AOT mainly based on the macroscopy and histopathological appearance and not from the radiological appearance (refer table 1)

Please discuss that point

Reviewer 2 Report

In the manuscript titled “Clinico-pathological presentations of cystic and classic adenomatoid odontogenic tumours”, the authors presented clinical and pathological features of the two variants of AOT and made an effort to identify the differences between them.

Following are my comments and suggestions:

Was the diagnosis of cystic AOT based on both microscopic and macroscopic evidence of a cystic presentation or on either one of the two?

When discussing about the recurrence rates and treatment of AOT, I would suggest that treatment of each of the reported cases be included in Table 1.

To make a radiographic correlation, it may be worthwhile to include the radiograph of the aggressive lesion in a 15-year old, shown in Figure 3.

As the authors discussed the differential possibility of an adenoid ameloblastoma, was Ki-67 IHC performed on the lesion in Figure 3. If yes, what % was the proliferative activity? CK19 is a marker of odontogenic epithelium, and will be positive in both AOT and ameloblastoma.

In Table 2, the authors compared the published cases of cystic AOT (Srikant 2010 and Grover et al 2015) with cases reported in the present study. I did not find the Grover et al publication in the references.

The authors performed a comparison between classic AOT, hybrid AOT and CEOT tumors and classic CEOTs. Were the hybrid and classic CEOT tumors from their biopsy service? This should be written in the Materials and Methods. There is no mention of these either in this section or in the abstract.

Author Response

2 Comments and Suggestions for Authors

In the manuscript titled “Clinico-pathological presentations of cystic and classic adenomatoid odontogenic tumours”, the authors presented clinical and pathological features of the two variants of AOT and made an effort to identify the differences between them.

Following are my comments and suggestions:

Was the diagnosis of cystic AOT based on both microscopic and macroscopic evidence of a cystic presentation or on either one of the two?

ANSWER

The cystic AOT was diagnosed based on both macroscopic and microscopic presentation  (please Refer Materials and method section ).

When discussing about the recurrence rates and treatment of AOT, I would suggest that treatment of each of the reported cases be included in Table 1.

ANSWER

Majority of the AOTs were enucleated followed by curettage of bone except for three lesions which received radical surgical excision. This fact will be mentioned in table 1 as follows

Type of surgery

Enucleation

10 (90.9)

28 (93.3)

38 (92.7)

Radical surgery

01 (03.3)

02 (06.7)

03 (07.3)

To make a radiographic correlation, it may be worthwhile to include the radiograph of the aggressive lesion in a 15-year old, shown in Figure 3.

ANSWER

As the radiographs are given to the patient, it is not possible include it as after several attempts it was not possible to trace the patient.

As the authors discussed the differential possibility of an adenoid ameloblastoma, was Ki-67 IHC performed on the lesion in Figure 3. If yes, what % was the proliferative activity?

ANSWER

We did not perform Ki-67 IHC to assess the proliferation activity.

CK19 is a marker of odontogenic epithelium, and will be positive in both AOT and ameloblastoma.

ANSWER

Agree, CK 19 cannot be used to differentiate Adenoid ameloblastoma from AOT or ameloblastoma as CK 19 will give positive reaction for all 3 tumours

In Table 2, the authors compared the published cases of cystic AOT (Srikant 2010 and Grover et al 2015) with cases reported in the present study. I did not find the Grover et al publication in the references.

ANSWER

Thank you for pointing out our mistake. It will be added to the reference list by removing the letter to the editor as both current  ref  5 and 6 deals with the same group of lesions as follows

6. Srikant N. Letter to the editor: Cystic adenomatoid odontogenic tumour: the master of disguise. Int J PediatrOtorhinolaryngol 2010; 74 : 836-7. Grover S, Rahim AMB, Parakkat NK, Kapoor S, Mittal K, Sharma B, Shivappa AB. Cystic odontogenic   Tumour. Case reports in Dent 2015: 503059; PMID 26579317

The authors performed a comparison between classic AOT, hybrid AOT and CEOT tumors and classic CEOTs. Were the hybrid and classic CEOT tumors from their biopsy service? This should be written in the Materials and Methods. There is no mention of these either in this section or in the abstract.

ANSWER

Yes both were from the same biopsy service. This fact will be mentioned in the materials and method section as follows

In addition, 09 CEOTs present in the archives of the Department of Oral Pathology were included for the comparison between classic AOTs, AOT containing CEOT like areas and CEOT.

Reviewer 3 Report

the describes small but inteting study, on this rare topic.

It carries important message worth of publication.

English revision required.

Author Response

3 Comments and Suggestions for Authors

They describes small but interesting study, on this rare topic.

It carries important message worth of publication.

English revision required.

Thank you very much for your kind comments

Reviewer 4 Report

Manuscript title: “Clinico-pathological presentations of cystic and classic adenomatoid odontogenic tumours”.

This is an interesting work evaluating the clinicopathological features of cystic and conventional variants of Adenomatoid odontogenic tumour (AOT), a tumour characterized by a unique biological profile. In particular, the Authors supported the hypothesis of cystic AOTs derived from dentigerous cysts. Nevertheless, no significant differences were found between these two subgroups.

Although the sample studied was small, this study represents an interesting starting point for discussion on the nature of odontogenic lesions.

However, there are some minor concerns to revise that are described below:

Page 3 line 92: delete “X2 test” in the round brackets. Page 3 line 95: in case of p<0.05 it is better to report the precise value of p. I think that should be necessary to improve readability of Tables. In particular: Table 1: I suggest to delete the percentage values in the round brackets. In case of numbers <10, report these numbers not preceded by zeroes. The note at the bottom of the table is not necessary for understanding. Tables 2 and 3: I suggest to be more concise. Page 10 lines 207-212: the Authors reported the higher frequency of dentigerous cyst in paediatric male patients as a evidence in favour to convince the fact that cystic AOTs derived from dentigerous cysts are more common in males. As the importance of the topic, I suggest to update the literature by citing the work of Lo Muzio et al. [1].

[1]: Lo Muzio L, et al. Cystic lesions of the jaws: a retrospective clinicopathologic study of 2030 cases. Oral Surg Oral Med Oral Pathol Oral Radiol. 2017;124:128-38.

Author Response

4 This is an interesting work evaluating the clinicopathological features of cystic and conventional variants of Adenomatoid odontogenic tumour (AOT), a tumour characterized by a unique biological profile. In particular, the Authors supported the hypothesis of cystic AOTs derived from dentigerous cysts. Nevertheless, no significant differences were found between these two subgroups.

Although the sample studied was small, this study represents an interesting starting point for discussion on the nature of odontogenic lesions.

However, there are some minor concerns to revise that are described below:

Page 3 line 92: delete “X2 test” in the round brackets. It was deleted

Page 3 line 95: in case of p<0.05 it is better to report the precise value of p. I think that should be necessary to improve readability of Tables.

ANSWER

Exact p values were added in the text as well as table 1

In particular: Table 1: I suggest to delete the percentage values in the round brackets. In case of numbers <10, report these numbers not preceded by zeroes. The note at the bottom of the table is not necessary for understanding.

Answer

Changes were made as requested

Tables 2 and 3: I suggest to be more concise.

Answer

As all information included in table 2 and 3 are relevant to the topic discussed, authors feel that the tables should remain as it is. However, if the reviewer can suggest a way to make it more concise without compromising the data, authors are willing to do so

 Page 10 lines 207-212: the Authors reported the higher frequency of dentigerous cyst in paediatric male patients as an evidence in favour to convince the fact that cystic AOTs derived from dentigerous cysts are more common in males. As the importance of the topic, I suggest to update the literature by citing the work of Lo Muzio et al. [1].

ANSWER

As requested ref 13 was deleted and the current ref by Muzio et al was included.

Line 210 /209 of page 10 was changed as follows

According to Lo Muzio et al 2017 [13], analyzing 152 dentigerous cysts in a pediatric population reveals a male to female ratio of 1.1:1. 

Ref 13 was changed to: Lo Muzio L, et al. Cystic lesions of the jaws: a retrospective clinicopathologic study of 2030 cases. Oral Surg Oral Med Oral Pathol Oral Radiol. 2017;124:128-38.